∂ | Open Peer Review | Environmental Microbiology | Research Article

# Unfolding the secrets of microbiome (Symbiodiniaceae and bacteria) in cold-water coral

Sanqiang Gong,[1,2] Jiayuan Liang,[2] Xujie Jin,[1] Lijia Xu,[3] Meixia Zhao,[1] Kefu Yu[2]

**ABSTRACT**    Recent deep-ocean exploration has uncovered a variety of cold-water coral (CWC) ecosystems around the world ocean, but it remains unclear how microbiome is associated with these corals at a molecular levels. This study utilized metabarcoding, tissue section observation, and metatranscriptomes to investigate the microbiome (Symbiodiniaceae and bacteria) of CWC species (*Narella versluysi*, *Heterogorgia uatumani*, and *Muriceides* sp.) from depths ranging from 260 m to 370 m. Warm-water coral (WWC) species (*Acropora pruinosa*, *Pocillopora damicornis*, and *Galaxea fascicularis*) were used as control groups. Results revealed that CWC host diverse bacteria and Symbiodiniaceae cells were observed in endoderm of CWC tissues. Several new candidate bacterial phyla were found in both CWC and WWC, including Coralsanbacteria, Coralqiangbacteria, Coralgsqaceae, Coralgongineae, etc. Both the 16S rRNA gene sequencing and metatranscriptomes revealed that Actinobacteria and Proteobacteria were abundant bacterial phyla in CWC. At the gene transcription level, the CWC-associated Symbiodiniaceae community showed a low-level transcription of genes involved in photosynthesis, $CO_2$ fixation, glycolysis, citric acid cycle, while bacteria associated with CWC exhibited a high-level transcription of genes for carbon fixation via the Wood-Lijungdahl pathway, short chain fatty acids production, nitrogen, and sulfur cycles.

**IMPORTANCE**    This study shed new light on the functions of both Symbiodiniaceae and bacteria in cold-water coral (CWC). The results demonstrated that Symbiodiniaceae can survive and actively transcribe genes in CWC, suggesting a possible symbiotic or parasitic relationship with the host. This study also revealed complete non-photosynthetic $CO_2$ fixation pathway of bacteria in CWC, as well as their roles in short chain fatty acids production and assimilation of host-derived organic nitrogen and sulfur. These findings highlight the important role of bacteria in the carbon, nitrogen sulfur cycles in CWC, which were possibly crucial for CWC survival in in deep-water environments.

**KEYWORDS**    cold-water coral, microbiome, symbiosis, Symbiodiniaceae, bacteria

Cold-water coral (CWC) has been recognized since the 18th century (1). However, only recent advancements in acoustic survey techniques and submersible tools (such as Remotely-Operated Vehicle) have revealed the scale and abundance of CWC ecosystems around the world ocean (1, 2). CWC is a group of cnidarians that comprises stony corals (Hexacorallia, Scleractinia), soft corals (Octocorallia), black corals (Antipatharia), and hydrocorals (Stylasteridae) (1). CWC commonly thrives in aphotic environments and can be found at depths spanning 50-4000 m, with preferred temperatures ranging between 4℃ and 12℃ (1, 3, 4). Just like the well-known warm-water coral (WWC), researchers suggest that CWC could provide critical three-dimensional habitats, which produce biodiversity hotspots in the deep-waters (5–7). Additionally, CWC provides a haven for microbial associates (8, 9).

Address correspondence to Meixia Zhao, zhaomeix@scsio.ac.cn, or Kefu Yu, kefuyu@scsio.ac.cn.

The authors declare no conflict of interest.

See the funding table on p. 13.

Corals are complex entities that form intricate interactions with different groups of microorganisms, including Symbiodiniaceae, fungi, bacteria, archaea, and viruses (10, 11). While the microbiome of the most commonly studied reef-building coral in warm-water habitats (WWC) has been extensively researched (10, 12–14), there has been limited attention given to the microbiome of CWC due to cost and difficulty in sample retrieval. Studies have shown that endosymbiotic Symbiodiniaceae supply food for WWC via photosynthesis (15, 16). Bacterial communities associated with WWC has been proved to play several important roles, such as nitrogen fixation, sulfur cycling, antibiotic production (13, 17). Both Symbiodiniaceae and bacteria associated with WWC play active roles in the health and adaptive response of the host to environmental changes (18, 19).

The absence of photosynthetic Symbiodiniaceae in CWC has generally inferred, rather than empirically demonstrated (20, 21). As the light becomes attenuated in both intensity and width as depth increases, posing a major constraint on the photosynthesis of algae. However, recent studies have provided evidence to the contrary (20, 22, 23). In 2011, one study found Symbiodiniaceae in CWC (Antipatharians) living in greater depth (396 m) (20). Recently, Rouzé et al. reported on the deepest photosymbiotic stony coral (*Leptoseris hawaiiensis*) (172 m depth) and confirmed CWC host Symbiodiniaceae by next generation sequencing, predominantly of the genus *Cladocopium* (24). To better understand these inconsistencies in the literature, it is important to determine whether the Symbiodiniaceae cells can divide in CWC tissues, and the Symbiodiniaceae cells have metabolic and ecological functions.

Similar to WWC, CWC possesses a diverse and abundant bacterial community (25). Recent studies in 2006 have revealed the presence of host-specific bacteria in CWC that differed from the surrounding environment (9, 26). Similar results have been observed in stony corals *Leptoseris* spp. (8, 24, 25, 27–29) and *Madrepora oculata* (30) in deep waters (>54 m). While attention has been focused on *Leptoseris* spp., there is comparatively less knowledge on cold-water octocoral species (21, 31, 32). Despite the growing research, the specific functional roles of bacteria associated with CWC remain unclear at present.

In this study, we examined the microbiome (both Symbiodiniaceae and bacteria) using next generation sequencing of the ITS2 region of Symbiodiniaceae ribosomal RNA gene, bacterial 16S rRNA gene, tissue section observation, and metatranscriptomes of sampled CWC species (*Narella versluysi*, *Heterogorgia uatumani,* and *Muriceides* sp., Fig. 1) from depths ranging from 260 m to 370 m. Our present results indicated that Symbiodiniaceae can not only form valid associations with CWC but also survival with a low-level transcription of genes involved in core metabolic functions. We proposed that the composition and function of CWC-associated microbiome is subject to change in response to the unique environmental conditions of deep-water habitats that receive minimal surface irradiance (less than 1%).

## RESULTS

### Microbiome overviews

As obtaining replicate samples of each CWC species proved challenging in deep waters, our primary focus was on describing the different taxa of Symbiodiniaceae and bacteria found in sampled corals. Both CWC and WWC host Symbiodiniaceae of the *Cladocopium* and *Durusdinium* genera (Fig. 2A). At the genotype level, CWC main hosts types C1, C42u, and C-new (a new Symbiodiniaceae type) of the genus *Cladocopium* and WWC main hosts types C50c, D1, and D4 of the genuses *Cladocopium* and *Durusdinium* (Fig. 2B). The number of Symbiodiniaceae ITS2 sequences from CWC was generally low, except in the case of *N. versluys*i (Fig. 2C). Upon examining the tissue and cellular levels, we observed various Symbiodiniaceae entities (both coccoid and binary fission forms) in the endoderm tissues of CWC and WWC (Fig. 2E).

The bacterial phyla of Proteobacteria (29%–51%), Firmicutes (9%–45%), Acidobacteria (15%–18%), and Euryarchaeota (archaea, 2%–10%) were detected in both CWC and WWC (Fig. 3A). However, the bacterial phylum of Tenericutes was enriched in CWC (average

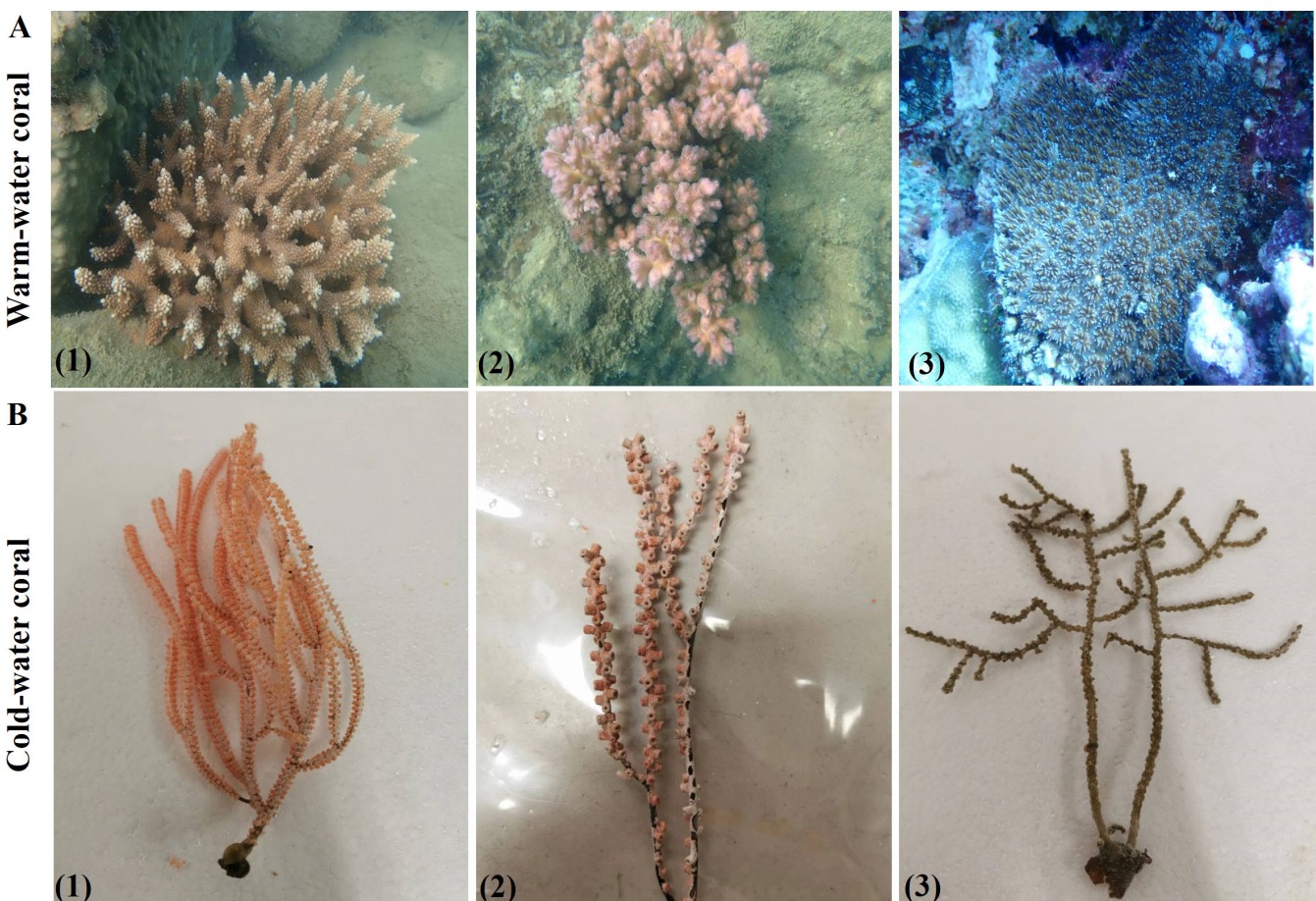

**FIG 1** Photographs of WWC (A) and CWC (B). (A) (1) *Acropora pruinosa*, (2) *Pocillopora damicornis*, (3) *Galaxea fascicularis*. (B) (1) *Narella versluysi,* (2) *Heterogorgia uatumani,* (3) *Muriceides* sp.

5%) (Fig. 3A). At fine scale taxonomic levels, various ASVs were identified as affiliated to different bacterial phylotypes, including *Streptococcus* spp., *Acetilactobacillus jinshanensis*, *Veillonella dispar*, *Fusobacterium canifelinu*, *Leptotrichia* spp., *Campylobacter* spp., *Prevotella* spp., and *Actinomyces* spp., which were enriched in CWC (Fig. 3B). Moreover, new candidates of bacterial phylotypes were detected (Fig. 4; Fig. S1A through D: Phylogenetic trees of new candidate bacteria phylotypes, as shown in supplementary figures file), such as a phylum Candidatus Coralqiangbacteria, a genus Candidatus *Coralgsqia,* a species Candidatus *Leptotrichia coralgsqb* in CWC, a phylum Candidatus Coralsanbacteria, an order Candidatus Coralgsqaceae, and a suborder Candidatus Coralgongineae in WWC.

At gene transcription level, we observed that the differences of intergroup are higher than that of intragroup of CWCs and WWCs (Fig. 5). The total relative abundance of genes belonging to Symbiodiniaceae was more than 50% in WWC while that in CWC was less than 2%. Conversely, the total relative abundance of genes belonging to bacteria was more than 70% in CWC while that in WWC was less than 1% (Fig. 5A). The core metabolic pathways of Symbiodiniaceae (e.g., translation, transcription, energy metabolism, carbohydrate metabolism, lipid metabolism, amino acid metabolism, etc.) were detected in both CWC and WWC, although their metabolic levels (as indicated by gene transcriptional levels) in CWC were low. On the contrary, the metabolic levels of bacterial pathways were high in CWC (Fig. 5C). We also found actively transcribed genes of hosts, which variation trend was similar with that of Symbiodiniaceae (Fig. 5C; Additional file 1: list of actively transcribed genes of host).

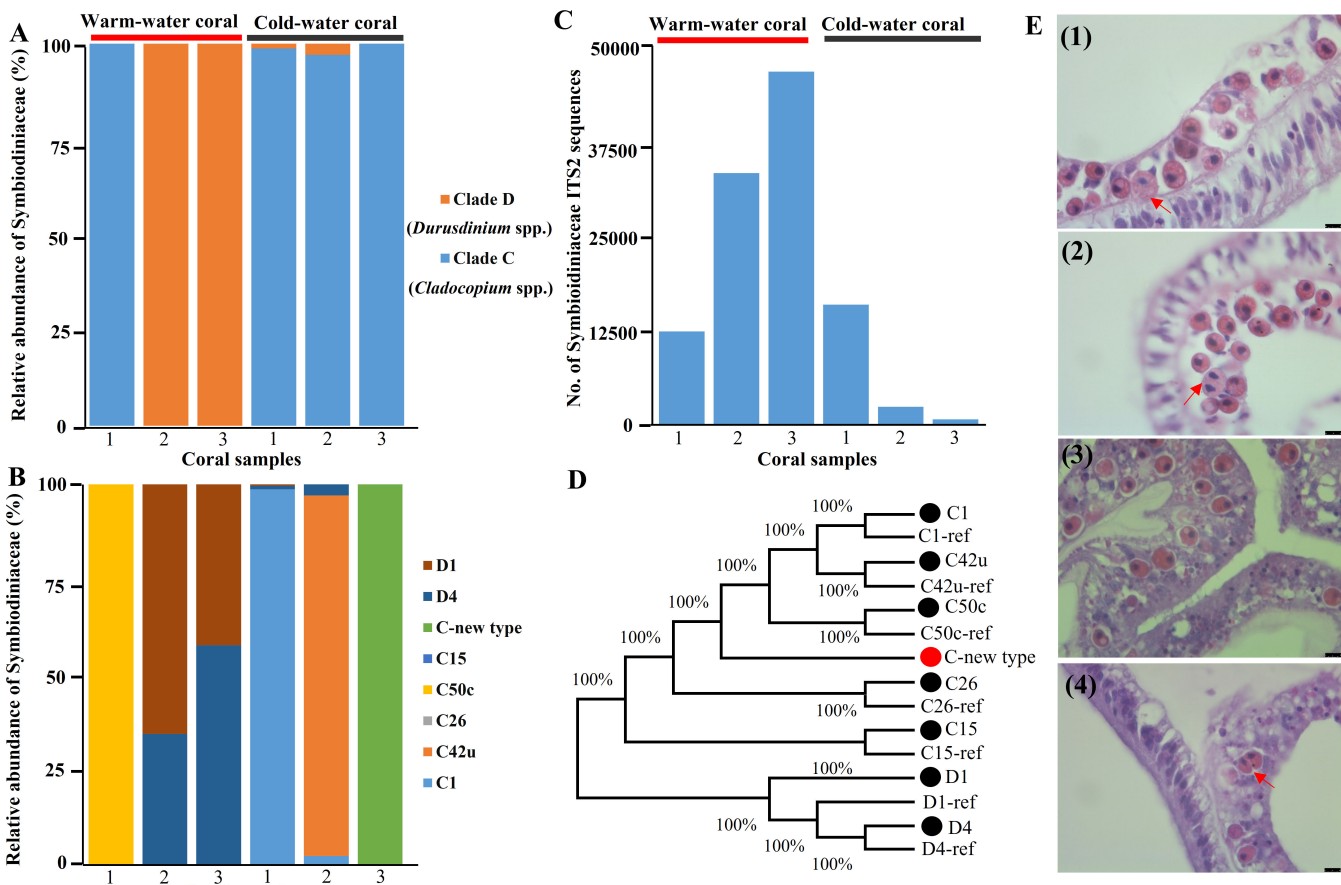

**FIG 2** Profiles of Symbiodiniaceae detected in explored WWC and CWC samples. Bar plot of Symbiodiniaceae communities in explored WWC and CWC (A, genus level, and B, type level). Total abundance of Symbiodiniaceae ITS2 sequences from WWCs and CWC (C). Phylogeny of the Symbiodiniaceae types based on ITS2 sequences from explored WWC and CWC values at nodes represent Bayesian posterior probabilities and ML bootstrap values (D). Optical images of tissue sections of WWC (1, 2) and CWC (3, 4). Arrows (red) represent Symbiodiniaceae cells in binary fission form (E).

## Gene functions of symbiodiniaceae

Our observations revealed that Symbiodiniaceae, associated with both CWC and WWC, actively transcribed genes related to light-driven energy transformation, $CO_2$ fixation via photosynthesis, glycolysis, citric acid cycle, and other core functional pathways (Fig. 6; Additional file 2: list of actively transcribed genes of Symbiodiniaceae). For instance, *LHCA1* and *LHCB1* genes involved in light-harvesting complex (LHC) protein; *psaA*, *psaB*, *psaC*, *psaD*, *psaE,* and *psaF* genes involved in photosystem I; *psbA*, *psbB*, *psbC*, *psbD*, *psbF*, *psbJ*, *psbK*, *psbL*, *psbO*, *psbF*, *psbU*, and *psbV* genes involved in photosystem II; and *ATPeF1A* gene involved in photosynthetic ATP synthesis were actively transcribed in CWC. Similarly, genes of *rbcL*, *pgk*, *gapA*, *fbaA*, *tktB*, *sbp*, *rpiA,* and *prkB* involved in photosynthetic $CO_2$ fixation via the calvin cycle were actively transcribed in CWC as well. Furthermore, genes of *HK*, *pfkA*, *pgi, fad*, *tpiA*, *gpmA*, *eno*, *aceE*, *gltA*, *SDH1*, and *MDH2* involved in the glycolysis and citric acid cycle were also actively transcribed in CWC. In addition, genes encoding proteins related to inorganic N/P assimilation, such as *nrtP*, *amt*, *nr*, *nirB*, *glnA,* and *pho*, were actively transcribed in CWC. However, we observed transcriptional levels of these genes was low in Symbiodiniaceae associated with CWC (Fig. 6).

## Gene functions of bacteria-carbohydrates utilization and metabolisms

Bacteria associated with CWC were found to have a notable role in carbohydrate utilization and metabolism as evidenced by actively transcription of numerous genes related to carbohydrate import, cellulose, and starch degradation (Fig. 7 and 8), such as

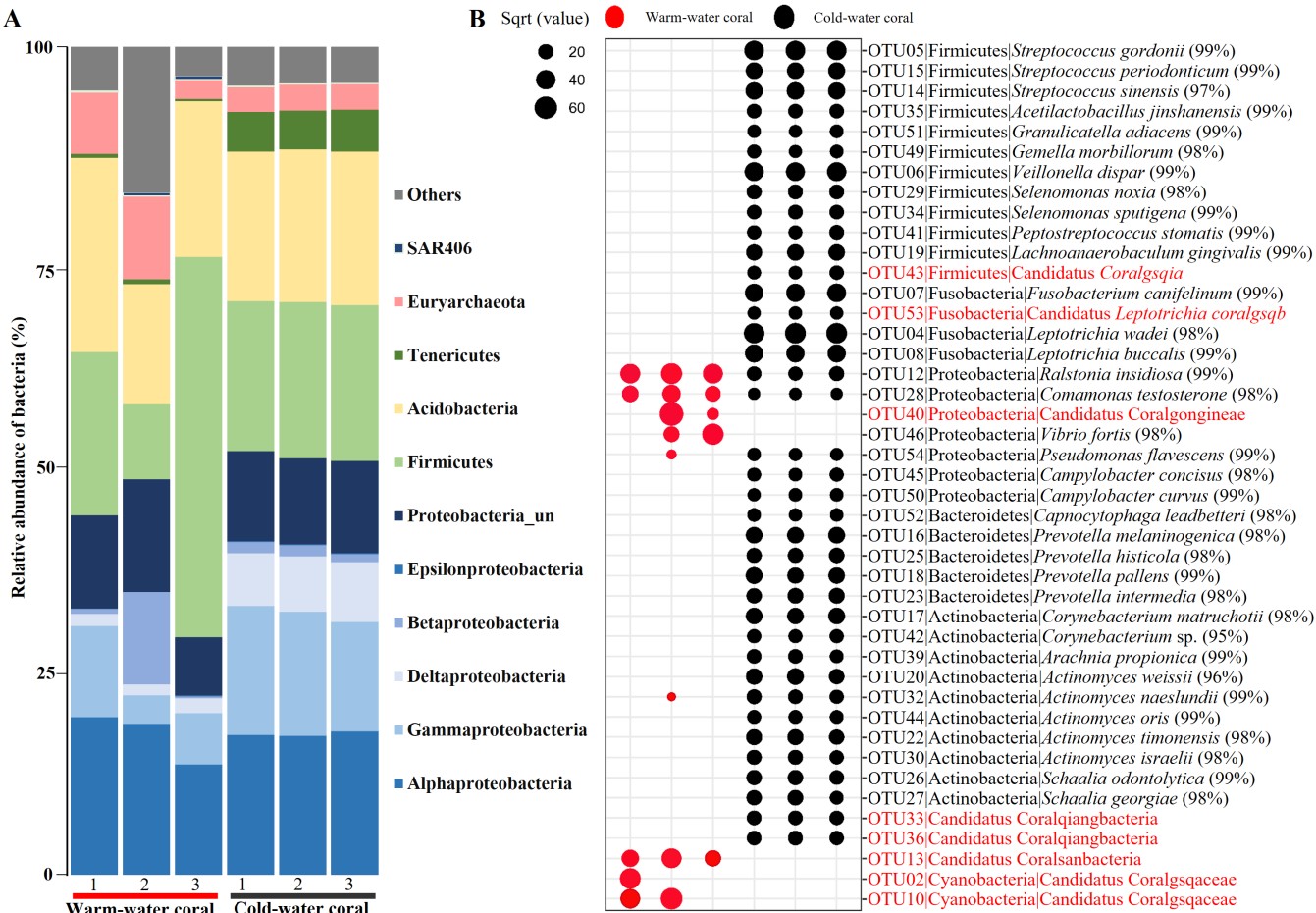

**FIG 3** Profiles of bacteria detected in explored WWC and CWC samples. Bar plot of bacteria communities in explored WWC and CWC (A). Profiles of most abundant bacteria (top 43) detected in explored coral samples. Relative abundance was sqrt transformed for plotting. The top panel represents coral species of WWC and CWC. The right panel shows the taxonomic affiliations of the bacteria (B).

ABC-2 (551 times), *bcsZ* (10 times), *malL* (6 times), *malZ* (210 times), *bglX* (9 times), *bglB* (11 times), *bglA* (64 times), *amyA* (62 times), *pulA* (>2 times), *treX* (2,650 times), *glgX* (>5 times), and *SGA1* (>2 times) genes (Fig. 7A). The relative abundance of these genes exhibited an increase in CWC as compared to that in WWC. Importantly, there was an increased activity in $CO_2$ fixation via the Wood-Lijungdahl pathway in bacteria associated with CWC. A 58 to 1,786 times increase in the relative abundance of genes encoding enzymes of the Wood-Lijungdahl pathway was detected, including *fdhF* (466 times), *fhs* (58 times,), *folD* (429 times), *metF* (267 times), *metE* (232 times), *metH* (1786 times), *acsS* (317 times), *acsA* (70 times), *pta* (70 times), and *ackA* (69 times) genes.

The bacteria associated with CWC were observed to have an increased capacity for the production of short chain fatty acids, including acetate, butyrate, and propionate (Fig. 7 and 8). Genes encoding enzymes involved in the production of these short chain fatty acids were found to be 16 to 4,195 times more abundant in CWC. The acetate production from acetyl-CoA was aided by *pta* and *ackA* genes, the butyrate production from acetyl-CoA was facilitated by *atoB* (897 times), *fadB* (203 times), *crt* (4,195 times), *ccrA* (96 times) genes, the propionate production from pyruvate involved *ppc* (417 times), *mdh* (22 times), *fumA* (57 times), *fumB* (168 times), *frdA* (95 times), *frdB* (17 times), *frdC* (26 times), *sucC* (795 times), *sucD* (4,578 times), *MUT* (>43 times), *epi* (>16 times), and *mlycd* (36 times) genes. These genes were almost all assigned to the Actinobacteria and Proteobacteria (Additional file 3: list of actively transcribed genes of bacteria).

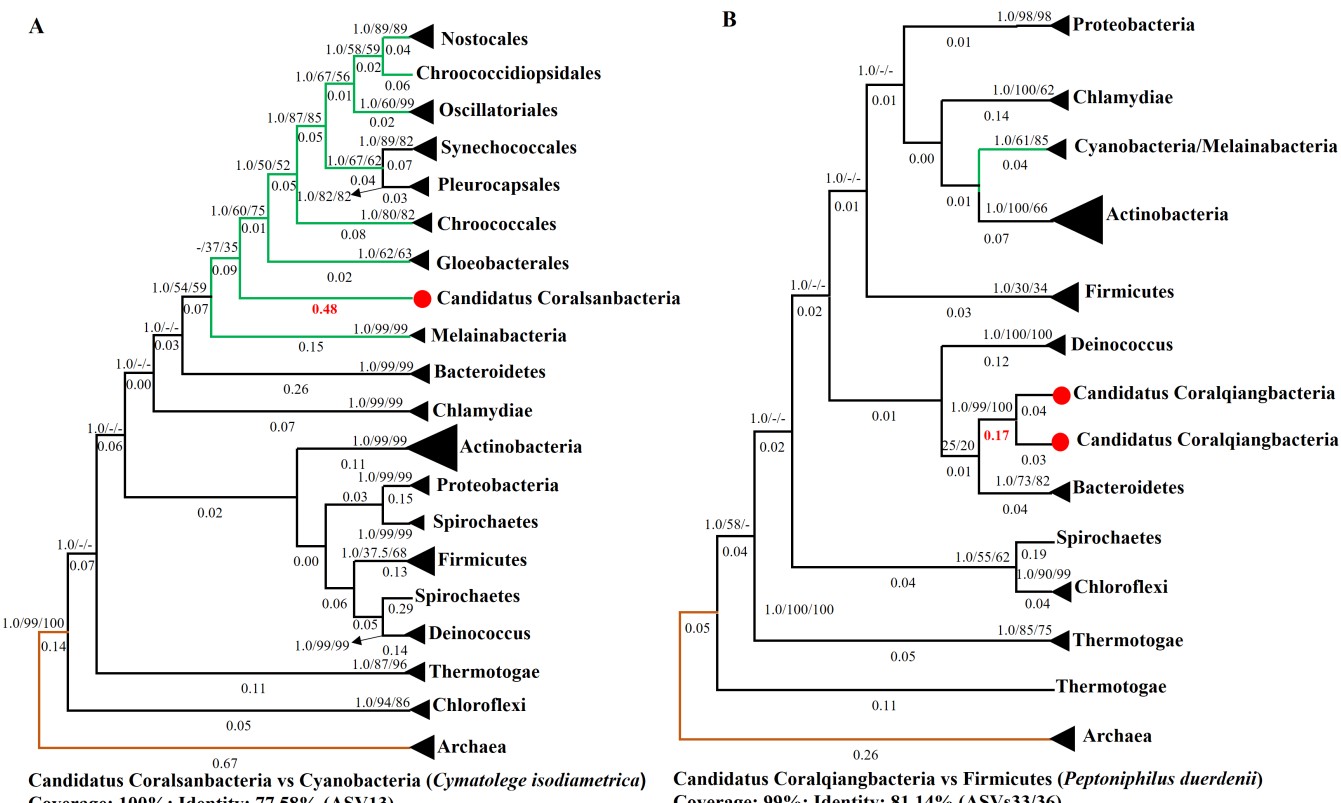

**FIG 4** Phylogeny of two new candidate bacteria phyla. Phylogenetic trees based on 16S rRNA gene sequences derived from our work and Rienzi's study (33) Support values at the nodes represent Bayesian posterior probabilities, ML/NJ bootstrap support. Phylogenetic analysis showing the Candidatus Coralsanbacteria is located at the bottom of Cyanobacteria (this tree including seven orders of cyanobacteria and melainabacteria) (A). The Candidatus Coralqiangbacteria is located near to Bacteroidetes and Deinococcus (B).

## Gene functions of bacteria-nitrogen cycle

Bacteria associated with CWC were found to have an increased potential for nitrogen cycle processes, including nitrogen transport, inorganic nitrogen fixation, nitrification and denitrification (Fig. 7; Additional file 3: list of actively transcribed genes of bacteria). The relative abundance of three genes encoding nitrogen fixation protein (*nifB*, >0.5 times, *nifT*, >1 times, and *nifU*, 8 times) was higher in CWC-associated bacteria. Additionally, genes encoding proteins related to nitrification (*norB* and *norC*) and denitrification (*narG*, *narH*, *narI*, *nosZ*, etc.) were predominantly present in the Actinobacteria and Proteobacteria associated with CWC. Furthermore, the bacteria associated with CWC exhibited an escalated potential for urea transport and degradation, as well as glutamine and glutamate biosynthesis. This was evidenced by an increase in genes encoding urea transporter (63 times, *urtA/B/C/D*), urease subunits and accessory proteins (640 times on average), glutamine synthetase (1,518 times, *glnA*), and glutamate synthase (546 times, *gltB/D*) (Fig. 7).

## Gene functions of bacteria-sulfur cycle

Increased metabolic potential for organic sulfur dissimilation (taurine and aliphatic sulfonates) and inorganic sulfur (sulfate and thiosulfate) assimilation were detected in bacteria associated with CWC (Fig. 7C; Additional file 3: list of actively transcribed genes of bacteria). Specifically, the bacteria associated with CWC showed a significant increase in the relative abundance of genes responsible for the taurine degradation pathway. These genes included *tauA/B/C*, which transport taurine into cells and were increased by 23 times, and *tauD*, which degrades taurine to sulfite and was increased by 17 times. In addition, genes involved in the aliphatic sulfonate degradation pathway exhibited a

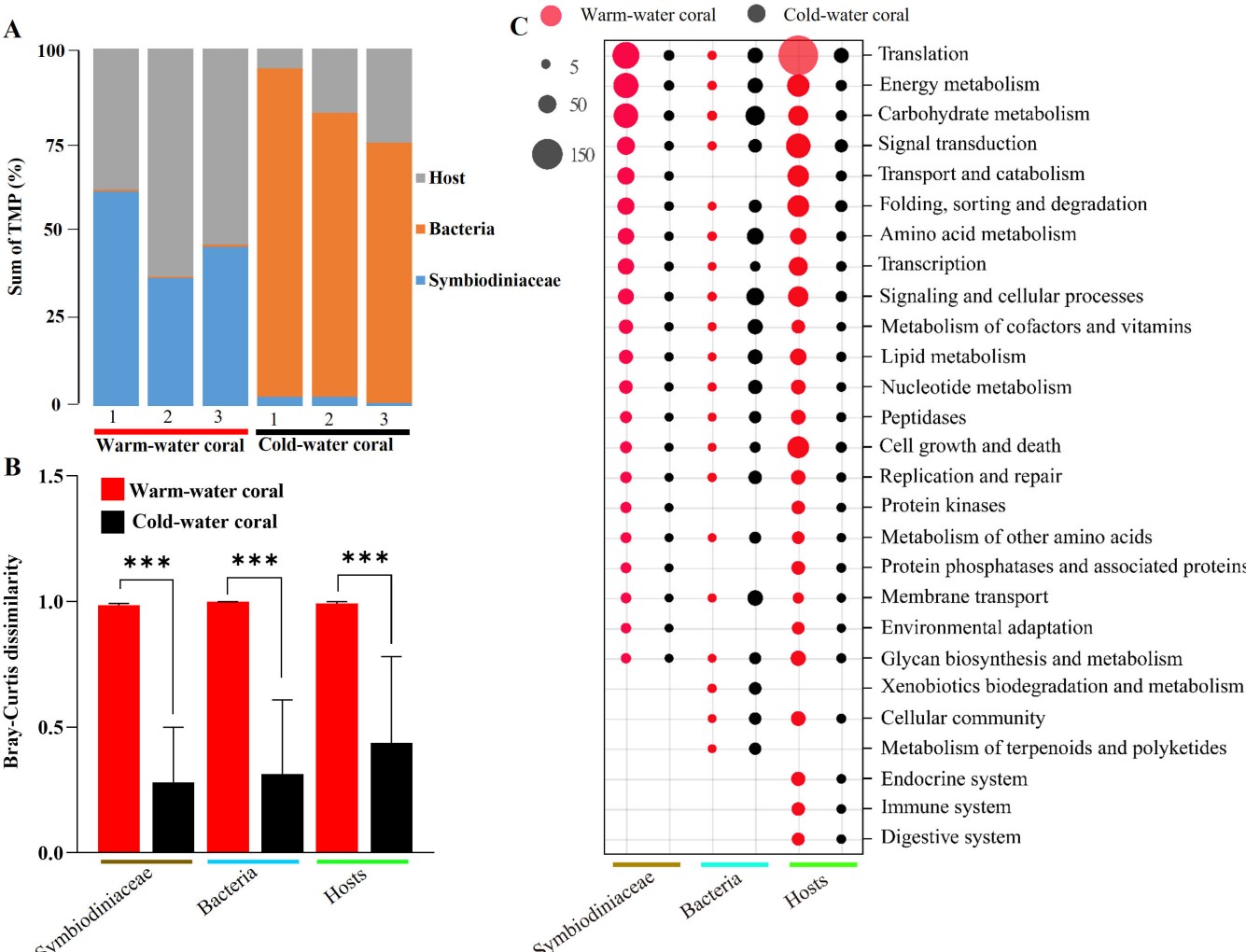

**FIG 5** Overall features of transcriptome profiles of WWC and CWC holobionts. Relative abundance of actively transcribed genes belongs to Symbiodiniaceae, bacteria, and hosts (A). Dissimilarities of gene transcription profiles of Symbiodiniaceae, bacteria, and hosts. Dissimilarities were calculated as Bray Curtis distances (B). Relative abundance of the most abundant pathways for Symbiodiniaceae, bacteria, and hosts based on actively transcribed genes (C). Mean relative abundances ($n$ = 3) were summed per KEGG pathway. Mean relative abundances of all genes belonging to each of the pathways were summed regardless of $P$-value.

notable increase in CWC-associated bacteria, including *ssuA/B/C/D/E* (>8 times). Furthermore, these bacteria exhibited a greater potential for inorganic sulfur assimilation, as evidenced by the presence of genes encoding enzymes for the conversion of inorganic sulfur into L-cysteine. The relative abundance of related genes, such as *cysA*, *cysP*, *cysW*, *cysU*, *cysD*, *cysN*, *cysH*, *cysE*, *cysJ*, *cysG*, *cysP*, *cysM*, and *cysK*, was more abundant in bacteria associated with CWC (Fig. 7C). Taxa potentially contributing to these pathways include the phyla Actinobacteria and Proteobacteria (Additional file 3: list of actively transcribed genes of bacteria). Overall, these findings suggest that deep-water environments thus enhance the ability of bacteria associated with CWC to utilize taurine and inorganic sulfur for biomass assimilation.

## DISCUSSION

### Symbiodiniaceae in CWC

As depth increases, the solar radiation becomes attenuated in both intensity and width, which limiting the growth of photosynthetic organisms (1, 3, 4). Thus, the CWC living in

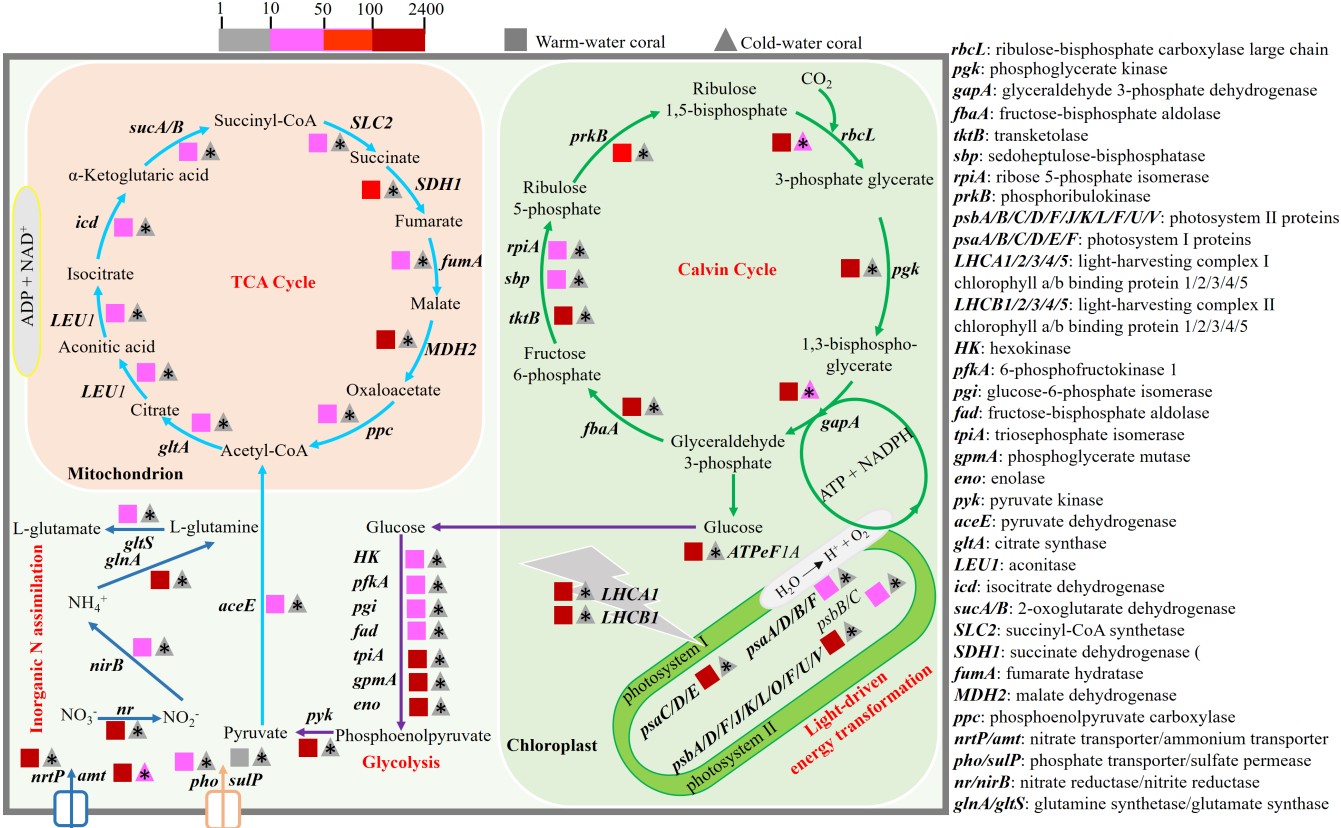

**FIG 6** Symbiodiniaceae metabolic pathways related to light-driven energy transformation, CO₂ fixation, glycolysis, citric acid cycle, and N/P metabolisms in WWC and CWC. Line color represents different metabolic pathways and colored circles represent ratios of relative abundances, all according to the scale described in the figure. *P*-values (represented by asterisks) were derived from ANOVA.

deep waters is commonly known as azooxanthellate over a long period of time (20, 21, 34). In 2011, Wagner et al. reported that most Hawaiian CWC (black corals) living in depth down to 396 m contains Symbiodiniaceae (20). Recent studies also identified Symbiodiniaceae in *Leptoseris* spp. (cold-water stone corals) down to 70 m in the Great Barrier Reef (23), down to 125 m in Hawaii, and down to 172 m in the Gambier archipelago (35). In the present study, we observed that CWC living depth down to 300 m in the northern of South China Sea also contained Symbiodiniaceae. Similarly, our present result revealed that these explored CWC host *Cladocopium* and *Durusdinium*, which representing two major groups of Symbiodiniaceae in WWC and CWC of the Indian and Pacific Oceans (12, 36). Importantly, we reported binary fission cells of Symbiodiniaceae in the endoderm tissues of the CWC living in deep waters for the first time. In addition, we firstly detected active transcription of nearly all genes related to photosynthesis, CO₂ fixation, glycolysis, citric acid cycle, and other core function pathways of Symbiodiniaceae in the explored CWC. These results demonstrated that Symbiodiniaceae can survive and actively transcribe genes in CWC, suggesting a possible symbiotic or parasitic relationship with the host.

## Bacteria in CWC

The present results revealed that the bacteria associated with CWC had an enhanced functional potential for energy-efficient carbon, nitrogen, and sulfur metabolisms. Unlike WWC that mainly harbor photosynthetic Symbiodiniaceae to supply the coral carbon requirements by the Calvin cycle (14), the present results indicated that bacteria associated with CWC displayed enhanced gene transcription of the Wood-Lijungdahl

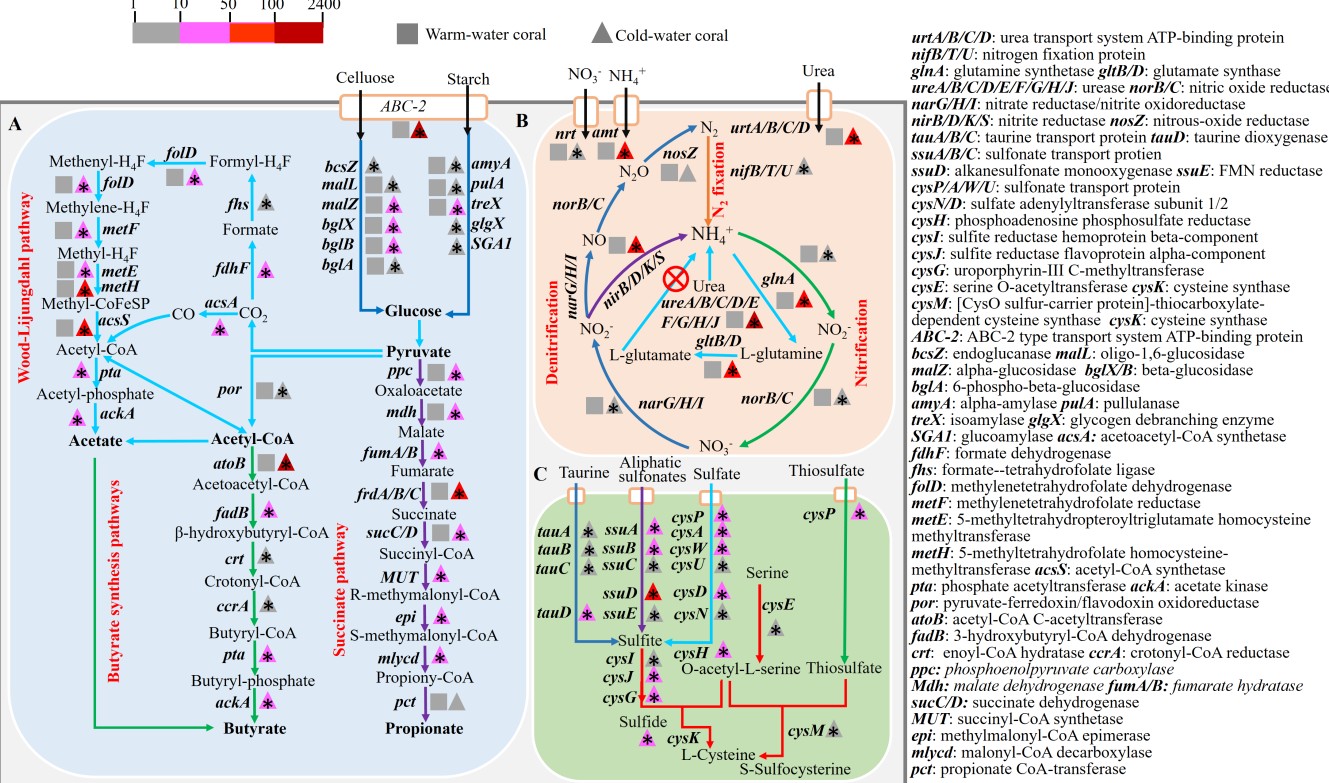

**FIG 7** Bacteria metabolic pathways related to carbon, nitrogen, and sulfur metabolism in WWC and CWC. Genes involved in transportation and degradation of exogenous carbon resources, $CO_2$ fixation, and short chain fatty acids (SCFAs) production (A). Genes involved in degradation and utilization of inorganic and organic nitrogen (B). Genes involved in inorganic sulfur assimilation and organic sulfur disassimilation (C). *P*-values (represented by asterisks) were derived from ANOVA. Line color represents different metabolic pathways and colored circles represent ratios of relative abundances, all according to the scale described in the figure.

pathway (e.g., *fdhF*, *fhs*, *folD*, *metF*, *metE*, *metH,* and *acsS*) for $CO_2$ fixation. The Wood-Ljungdahl pathway is the largest $CO_2$ fixation pathway in anaerobic conditions by chemoautotrophic bacteria (37). This pathway differs from the Calvin-Benson cycle by its non-cyclic carbonic fixation that forms acetyl-CoA from $CO_2$, suggesting non-photosynthetic $CO_2$ fixation by bacteria is playing important roles in supplying organic carbon to CWC. Besides, the present results suggested that the organic carbon resource needed by CWC might be also acquired by exogenous carbohydrates (e.g., cellulose and starch). As cellulose and starch transport and degradation related genes (e.g., *ABC-2*, *bcsZ*, *malI*, *malZ*, *bglX*, *bglB*, *bglA*, *amyA*, *pulA*, *treX*, *glgX,* and *SGA1*) were actively transcribed in bacteria of CWC, which supports previous speculations that the CWC consume food for energy (38).

Furthermore, the actively transcription of genes related to short chain fatty acids synthesis (acetate, butyrate, and propionate) was firstly reported in the bacteria associated with CWC in our present study. In humans, short chain fatty acids are primarily derived from fermentation of dietary cellulose and play a pivotal role in regulating gut microbiota and host's metabolic and immune function (39). We, therefore, speculated that the enhanced metabolic penitential of SCFAs production by bacteria associated the CWC could be helpful in coral-microbiome symbiosis facing the deep waters characterized by dark and cold.

While it is evident that nitrogen metabolism is one of the key factors in the growth and health of WWC (40), little is known about nitrogen cycle in CWC. Herein, we detected actively transcription of genes encoding nitrogen fixation enzyme (*nifB*, *nifT*, and *nifU*) of bacteria (41), supporting the presence of nitrogen-fixating bacteria in CWC. Functional gene screening also revealed the present and transcription of nitrate reductase

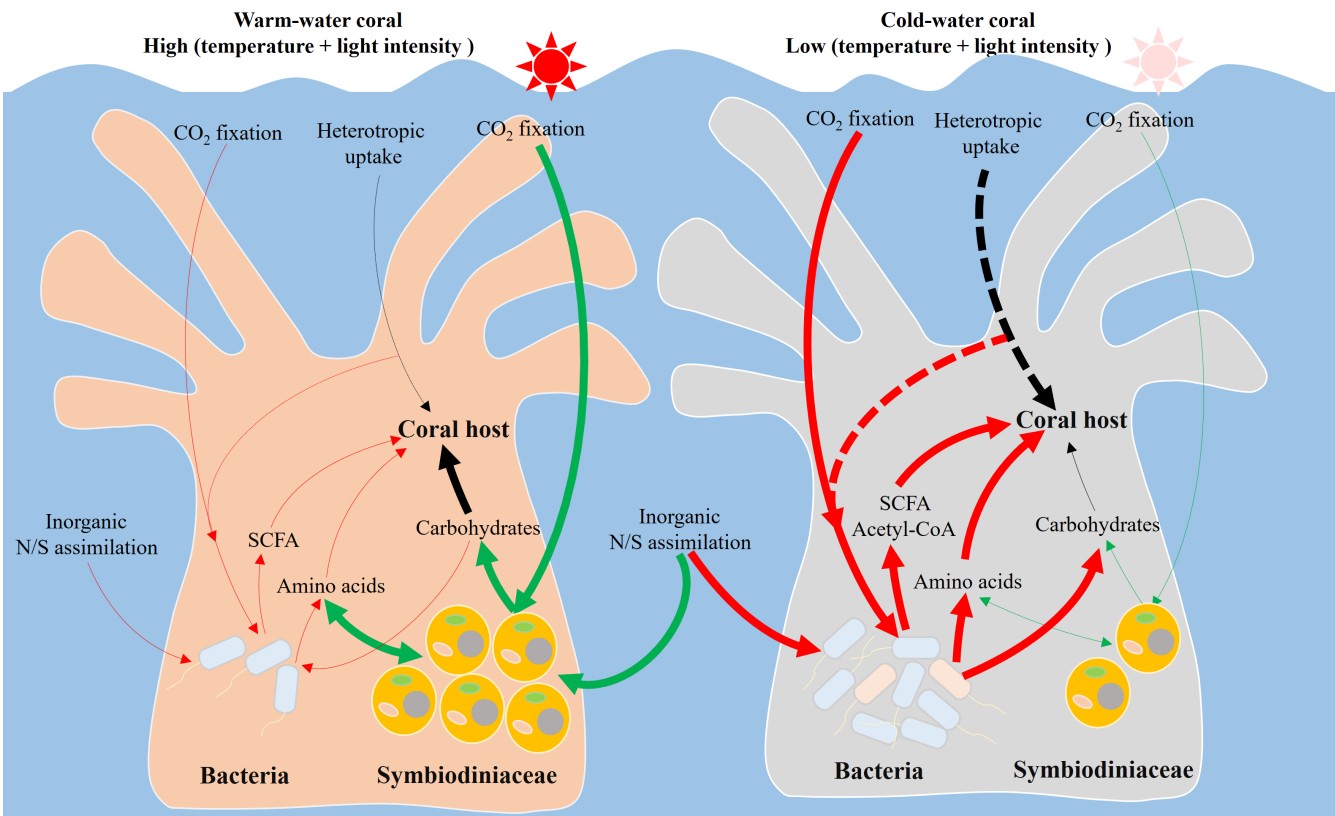

**FIG 8** Schematic summary of the symbiotic associations among Symbiodiniaceae, bacteria, and hosts in WWC (left) and CWC (right) holobionts. Line colors represent different metabolic pathways of Symbiodiniaceae (green), bacteria (red), and hosts (black), and line thicknesses represent relative abundances of different metabolic pathways.

(*narG/H/I*), nitrite reductase (*nirB/D/K/S*), nitrous-oxide reductase (*nosZ*), and nitric oxide reductase (*norB/C*) genes (42), indicating the potential for denitrification and nitrification pathways in bacteria of the CWC. Specifically, we detected an increase of genes for urea transport (*urtA/B/C/D*) and degradation (*ureA/B/C/D/E/F/G/H/J*) (43), as well as glutamine (*glnA*) and glutamate (*gltB/D*) biosynthesis (44). It, therefore, appears that the bacteria in the CWC have increased capacity to benefit from host-derived organic nitrogen metabolism. In all, the present results indicated that the bacteria associated with CWC contain members capable of numerous nitrogen transformation pathways, including nitrogen fixation, denitrification, nitrification, and organic nitrogen metabolism.

The present results also demonstrated widespread potential for sulfur metabolism in CWC. Taurine is a naturally occurring sulfur enriched organic compound in a wide of variety of marine organisms (45). The genes encoding encoding ABC transporters (*tauA/B/C*) and taurine dioxygenases (*tauD*) were detected in bacteria associated with CWC in our present study, which taking part in converting taurine to sulfite (46). In addition to the cleavage of taurine, sulfite may also originate from the aliphatic sulfonates and sulfate, as genes (*ssuA/B/C/D/E*) encoding enzymes for degradation of aliphatic sulfonates (47) and genes (e.g., *cysP/G/M/K*) encoding enzymes for reduction of inorganic sulfur (48) were enriched in bacteria associated with CWC. Host-derived taurine would provide a source of sulfur for the bacteria while organic sulfur synthesis from sulfide could balance sulfide generation due to sulfate reduction by bacteria, thus increasing the overall stability of sulfur cycle in CWC.

## Summary

In this study, the microbiome (Symbiodiniaceae and bacteria) associated with CWC from depth ranging from 200 to 300 m was investigated. More importantly, the

metabolite-enabled functions of microbiome were firstly described in CWC. The present results indicated that Symbiodiniaceae can survive and actively transcribe genes in CWC. The bacteria associated with CWC exhibited transcription of genes for the Wood-Lijung-dahl pathway, suggesting that the energy requirements of CWC were possibly acquired by bacteria via a non-photosynthetic $CO_2$ fixation pathway in deep water. Feeding under the help of bacteria and Symbiodiniaceae could be two alternative ways to the energy supply of CWC. The findings advanced general understanding of the mechanisms of CWC surviving in deep-water environments where receive less than 1% of surface irradiance. Furthermore, the bacteria has enhanced potential for short chain fatty acids production, providing strategies for regulating the function of CWC and its hosted microbiome. Meanwhile, the assimilation of host-derived organic nitrogen and sulfur through the roles of bacteria could increase the overall stability of the CWC holobiont. These features of community composition and function of microbiome are likely important for the efficient and successful survive of CWC at deep-water habitat.

## MATERIALS AND METHODS

### Sample collection

During a cruise in the eastern Hainan Island of the South China Sea (112.918–112.934° E, 18.741–18.764° N) on December of 2021, a total of three CWC samples (*N. versluysi*, *H. uatumani,* and *Muriceides* sp.) were collected by using a submersible (SHEN HAI YONG SHI). Stations for CWC sampling were at depths ranging from 260 m to 370 m (Additional file 4: physical and chemical conditions for each sampling site). For each CWC, the colony was pressing down on a plunger with the submersible's manipulator arm. Branches (about 5–10 cm in height) were collected from from CWC colonies using the submersible's manipulator claw. Branches of CWC were brought to the surface alive, washed three times with sterile seawater (10°C) and divided into fragments (approximately 1–3 cm$^2$) using sterile scissors. A total of three WWC samples (*Acropora* sp., *Galaxea fascicularis,* and *Platygyra lamellina*) were collected from, Sanya, Hainan Island, China (109.292° E, 18.123° N), a tropical coral reef, by snorkeling on December of 2021. Each WWC colony (about 10–12 cm in diameter) was divided into branches using sterile pincers. After washing with sterile seawater (22°C), the coral branch was further divided into fragments (approximately 1–3 cm$^2$) using sterile scissors. For each sampled coral species, several fragments ($n = 3–6$) were preserved in RNAhold (TRAN, China) and others ($n = 3–6$) were preserved in Sorensen-sucrose phosphate buffer (0.1 M phosphate at pH 7.5, 0.65 M sucrose, 2.5 mM $CaCl_2$) that containing 2.5% glutaraldehyde and 1% formaldehyde. All samples were stored at −20°C for further DNA/RNA extraction and/or tissue section observation.

All corals were identified based on morphology observation and molecular analysis of the mitochondrial *COI* gene (49) (Fig. 1; Additional file 5: sequences of *COI* gene).

Metadata including location, depth, temperature, and salinity were recorded (Additional file 4: physical and chemical conditions for each sample site). CWC were sampled from depths ranging from 260 m to 370 m, where the average temperature and salinity were 10°C and 33.4 ‰, respectively. WWC were sampled from depths ranging from 3 m to 6 m, where the average temperature and salinity are 23°C and 35.2‰, respectively.

### Tissue section observations

For tissue section observations, the fixed coral fragments were decalcified according to Kopp's method (50). In detail, the fixed coral fragments were decalcified at 4°C in Sorensen-sucrose phosphate buffer containing 0.5 M EDTA. The decalcification buffer was renewed daily until being completely demineralized. Rinsed coral tissue samples were dissected and post-fixed in 1% $OsO_4$ in Sorensen-sucrose phosphate buffer for

1 hour at room temperature, and then dehydrated in ethanol and embedded in Spurr resin. Sections were cut with a Diamote 35° diamond (Ultracut microtome). Tissue sections were stained with HE (hematoxylin-eosin staining) and observed with a light microscope LEICA DMRB equipped with a LEICI DC300F camera (Leica, France).

## DNA extraction, amplification, pyrosequencing, and data processing

Total genomic DNA of coral samples was extracted using a Qiagen DNeasy Kit (Qia-gen, Hilden, Germany) according to the manufacturer's protocol. In order to identify Symbiodiniaceae and bacteria community compositions in corals, the ITS2 region of Symbiodiniaceae ribosomal RNA gene was PCR amplified with ITS2 primers of ITSintfor2 (5′-GAATTGCAGAACTCCGTG-3′) and ITS2-reverse (5′GGGATCCATATGCTTAAG-TTCAGCGGGT-3′) (12) and the V3 and V4 hypervariable region of bacterial 16S rRNA genes were amplified with the primers 341F (5′-CCTAYGGGRBGCASCAG-3′) and 806R (5′-GGACTACNNGGGTATCTAAT-3′) (13). After pooling multiple samples in one run of Illumina sequencing (MiSeq), a unique 12-mer tag for each DNA sample was added to the 5′ end of primers. Each sample was PCR-amplified in a 50 µL reaction, which contained 25 µL Multiplex Taq (Qiagen, Hilden, Germany), 10 mM of each primer, 60 ng of genomic DNA, and DNase-free water to a total volume of 50 µL. Cycling conditions were set as: 94°C for 5 minutes followed by 30 cycles of denaturation at 94°C for 30 seconds, annealing at 52°C for 30 seconds, extension at 72°C for 30 seconds, and a final extension at 72°C for 10 minutes. The PCR products were validated by Agilent 2100 Bioanalyzer (Agilent Technologies, Palo Alto, CA, USA), and quantified by Qubit 3.0 Fluorometer (Life Technologies, New York, NY, USA). Finally, the PCR products were performed using a $2 \times 300$ paired-end configuration. Base calling was done by the MiSeq Control Software embedded in the Illumina MiSeq instrument.

Raw reads of the ITS2 region of ribosomal RNA gene of Symbiodiniaceae were analyzed with default settings using the SymPortal: a novel analytical framework and platform for coral-algal symbiont next-generation sequencing ITS2 profiling (51) and the 16S rRNA gene sequences were processed using QIIME2 (quantitative insights into microbial ecology) platform according to previous studies (13, 52).

## RNA extraction, sequencing, and metatranscriptomic analysis

Total RNA of coral samples was extracted using Qiagen RNeasy Kit (Qiagen, Hilden, Germany) according to the manufacturer's protocol. RNA quantity and integrity were analyzed using a NanoDrop ND-1000 spectrometer (Wilmington, DE, USA) and an Agilent 2100 Bioanalyzer (Santa Clara, CA, USA). RNA samples with high purity (OD260/280 between 1.9 and 2.1) and high integrity [RNA integrity number (RIN) >8.0] were used for further cDNA library construction. The sequencing were performed according to our previous study (53).

The quality control and analysis of raw reads were performed using SqueezeMeta software, a fully automatic pipeline for Metagenomic/metatranscriptomic analysis (54). In brief, The Trimmomatic-0.38 software was used for adapter removal, trimming, and filtering by quality according to default parameters (55). The obtained clean reads were further assembled using Megahit (56). The Diamond software (57) was used for homology searching of assembled gene sequences against several taxonomic and functional databases, including the eggNOG database (58), the latest publicly available version of KEGG database (59) and the PFAM database using HMMER3 (60) with default settings. For the taxonomic assignment, an LCA algorithm that looks for the last common ancestor of the hits for each query assembled gene using the results of the Diamond search against GenBank nr database (including coral and Symbiodiniaceae genomes) (54). To estimate the abundance of each assembled gene in each sample, original reads were mapped onto the contigs resulting from the assembly using the software Bowtie2 (61). RSEM software (62) was used to compute the average coverage and normalized TPM values that provide information on gene abundance.

## Statistics analyses

A non-metric multidimensional scaling analysis and a permutational multivariate analysis of variance using distance matrices were performed to test the significant differences of coral microbiome between different groups using the vegan package in the R software environment (R 3.1.2). The analysis of differential transcribed genes between WWC and CWC was performed using the DESeq2 method (63), with a threshold $P$-value of $< 0.05$, Fold Change $\geq 2$ and TPM (Transcripts Per Million) $\geq 3$. $P$ values were calculated using a two-sided ANOVA similar statistic test.

## ACKNOWLEDGMENTS

We thank submersible (SHEN HAI YONG SHI) for cold-water coral sampling.

This work was supported by the National Key R&D Program of China (Grant No. 2021YFC3100600), the Natural Science Foundation of Guangdong Province, China (2022A1515010521), the Opening Project of Guangxi Laboratory on the Study of Coral Reefs in the South China Sea, Nanning 530004, China (GXLSCRSCS2021002), and the Central Public-Interest Institution Basal Research Fund (PM-zx703-202105-176).

S.G.: Conceptualization, Investigation, Methodology, Formal analysis, Writing-original draft. Jiayuan Liang, Investigation, Writing-review and editing. X.J., Investigation, Writing-review and editing. L.X., Writing-review and editing. M.Z.: Conceptualization, Resources, Writing-review and editing. K.Y.: Resources, Writing-review and editing, and Funding acquisition. All authors read and approve the final manuscript.

## AUTHOR AFFILIATIONS

[1]Key Laboratory of Tropical Marine Bio-resources and Ecology & Guangdong Provincial Key Laboratory of Applied Marine Biology, South China Sea Institute of Oceanology, Chinese Academy of Sciences, Guangzhou, China
[2]Coral Reef Research Center of China, Guangxi University, Nanning, China
[3]South China Institute of Environmental Sciences, The Ministry of Ecology and Environment of PRC, Guangzhou, China

## AUTHOR ORCIDs

Sanqiang Gong http://orcid.org/0000-0002-4342-7712
Meixia Zhao http://orcid.org/0000-0002-7063-3108

## FUNDING

| Funder | Grant(s) | Author(s) |
| --- | --- | --- |
| MOST \| National Key Research and Development Program of China (NKPs) | 2021YFC3100600 | Meixia Zhao |
| Natural Science Foundation of Guangdong Province (Guangdong Natural Science Foundation) | 2022A1515010521 | Sanqiang Gong |
| Opening Project of Guangxi Laboratory on the Study of Coral Reefs in the South China Sea | GXLSCRSCS2021002 | Sanqiang Gong |
| Central Public-Interest Institution Basal Research Fund | PM-zx703-202105-176 | Lijia Xu |

## AUTHOR CONTRIBUTIONS

Sanqiang Gong, Conceptualization, Data curation, Formal analysis, Funding acquisition, Methodology, Visualization, Writing – original draft | Jiayuan Liang, Investigation, Methodology, Writing – review and editing | Xujie Jin, Funding acquisition, Investigation, Writing – review and editing | Lijia Xu, Conceptualization, Funding acquisition, Investigation, Writing – review and editing | Meixia Zhao, Conceptualization, Funding

acquisition, Investigation, Resources, Supervision, Writing – review and editing | Kefu Yu, Funding acquisition, Investigation, Resources, Supervision, Validation, Writing – review and editing

## DATA AVAILABILITY

The data generated as part of this study is controlled access. The raw sequence data (a total of six RNA sequencing libraries, six 16S rRNA sequencing libraries and six ITS2 sequencing libraries) produced in this study were deposited in the Sequence Read Archive (PRJNA897731 and PRJNA897088) of the NCBI (https://blast.ncbi.nlm.nih.gov). The source data underlying all figures are provided as supplementary data files.

## ADDITIONAL FILES

The following material is available online.

### Supplemental Material

**Legends of supplemental files (Spectrum01315-23-s0001.docx).** Descriptions of supplemental files 1 to 5.
**Fig. S1 (Spectrum01315-23-s0002.docx).** Phylogenetic trees of new candidate bacteria phylotypes.
**Supplemental file 1 (Spectrum01315-23-s0003.xlsx).** List of actively transcribed genes of host.
**Supplemental file 2 (Spectrum01315-23-s0004.xlsx).** List of actively transcribed genes of Symbiodiniaceae.
**Supplemental file 3 (Spectrum01315-23-s0005.xlsx).** List of actively transcribed genes of bacteria.
**Supplemental file 4 (Spectrum01315-23-s0006.xlsx).** Physical and chemical conditions for each sampling site.
**Supplemental file 5 (Spectrum01315-23-s0007.txt).** Sequences of *COI* gene.

### Open Peer Review

**PEER REVIEW HISTORY (review-history.pdf).** An accounting of the reviewer comments and feedback.

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
