## [Reviewer comments · Microbiology Spectrum]

Microbiology Spectrum

Unfolding the secrets of microbiome (Symbiodiniaceae and bacteria) in cold-water coral

Sanqiang Gong, Jiayuan Liang, Xujie Jin, Lijia Xu, Meixia Zhao, and Kefu Yu

Corresponding Author(s): Sanqiang Gong, South China Sea Institute of Oceanology, Chinese Academy of Sciences

Review Timeline:

Submission Date:	March 28, 2023
Editorial Decision:	April 28, 2023
Revision Received:	June 15, 2023
Accepted:	July 9, 2023

Editor: Konstantinos Kormas

Reviewer(s): Disclosure of reviewer identity is with reference to reviewer comments included in decision letter(s). The following individuals involved in review of your submission have agreed to reveal their identity: Jin Zhou (Reviewer #1); Helena Villela (Reviewer #2)

Transaction Report:

DOI: <https://doi.org/10.1128/spectrum.01315-23>

April 28, 2023

Dr. Sanqiang Gong
South China Sea Institute of Oceanology, Chinese Academy of Sciences
Guangzhou, China
Guangzhou
China

Re: Spectrum01315-23 (Unfolding the secrets of microbiome (Symbiodiniaceae and bacteria) in cold-water coral)

Dear Dr. Sanqiang Gong:

Link Not Available

Sincerely,

Konstantinos Kormas

Journals Department
Reviewer comments:

Reviewer #1 (Comments for the Author):

In this study, Gong et al. analyzed the microbiome of cold-water corals. they found that CWCs host low-abundance Symbiodiniaceae and predominantly of diverse communities of bacteria. Importantly, their present results indicated that Symbiodiniaceae might be a symbiont or parasite of CWCs and has cellular activity, these finds were important for photosynthetic algae to acclimation to deep water habitants. In addition, this study revealed that bacteria were abundant in CWCs, which exhibited a high-level transcription of genes for carbon fixation via the Wood-Ljungdahl pathway, as well as short chain fatty acids (acetate, butyrate and propionate) production, nitrogen and sulfur (especially host derived organic nitrogen and sulfur metabolisms) cycles, suggesting the energy requirements of CWCs were mainly acquired by bacteria via non-photosynthetic CO₂ fixation pathway. The findings advanced general understanding of the mechanisms of CWCs surviving in

deep-water environments, which which receive less than 1% of surface irradiance. In all, this is an interesting study. However, some minor revision is needed that may improve the quality of the manuscript.

1. Line 37: were should be revised to was
2. Line 41: CO₂ should be revised to CO₂.
3. Line 74: There is a problem of the reference format (Hourigan., 2007), please check and revise
4. Line 75: "been inferred" been should be deleted.
5. Line 76: There is a problem of the reference format (Hourigan., 2007), please check and revise
6. Line 77: for should be revised to on
7. Line 83: There is a problem of the reference format (Hourigan., 2007), please check and revise
8. Line 94: less is revised to less are
9. Line 156: Carbohydrates should be revised to carbohydrates
10. Line 178: Nitrogen cycle should be revised to nitrogen cycle
11. Line 179: of nitrogen cycle should be revised to for nitrogen cycle
12. Line 182: were primarily assigned should be revised to was primarily assigned
13. Line 186: an notable should be revised to a notable
14. Line 195: Sulfur cycle should be revised to sulfur cycle
15. Line 244-245: the sentence "indicating non-photosynthetic CO₂ fixation by bacteria is play important roles in organic carbon supply of CWCs." Should be revised to "indicating non-photosynthetic CO₂ fixation by bacteria is playing important roles in supplying organic carbon to CWCs."
16. Line 246: the word "acquire" should be revised to "acquired"
17. Line 251: the word "were" should be revised to "was"
18. Line 289: the word "were" should be revised to "was"
19. Line 330: In details should be revised to In detail
20. Line 378: the "a" should be revised to an
21. FIG 2. A and B cannot be found, please check.
22. FIG 3. A and B cannot be found, please check.

Reviewer #2 (Comments for the Author):

Major concerns:

- The paper is relevant to the field, showing important findings that will help to fill gaps in the area. However, the number of replicates (or even pseudoreplicates, it is still unclear) makes me think that the whole results and discussion sections will need to be reformulated. If the authors are using pseudoreplicates (as it seems like), they should not do any comparison and not work with abundance analyses but simply a description of the findings. Also, the statistics should be reviewed if this is the case. Once the authors make the methodology clear regarding the number of true replicates they used for each analysis, we can better judge what can be done to show and discuss the results.
- The paper needs English review AND technical review.

Abstract

- Line 20 - Change "habitants" for "habitats"? It is unclear.
- Line 20 - 21 - "Detected microbiome" does not sound right. Consider changing it to "In this study, we investigated the CWC-associated microbiome...".
- Line 22- The sentence is too long and has grammar mistakes. Consider changing it to "We also analyzed the metatranscriptomes of THE sampled CWCs...".
- Line 23 - "from depths down to 260-370 m" Is it the total range or the range of the deepest spot? It is confusing.
- Line 24 - It is plural, so it should be "were used as control groups".
- Line 25-26 - "and predominantly of diverse communities of bacteria" The sentence is confusing. Please rewrite it.
- Line 26-27 - "We designated several new candidate bacterial phyla" Grammar is wrong here and the word designated is not appropriate for what the authors mean.
- Line 28 - Remove "and/or", it should be "and" only.
- Line 29 - Consider changing "16S metabarcoding" to "the 16S rRNA gene sequencing".
- Line 30-32 - Consider changing it to "we noted that the coral-associated Symbiodiniaceae community showed a low-level transcription of genes involved in photosynthesis, CO₂ fixation, glycolysis, citric acid cycle and other core function pathways in the CWCs."
- Line 35 - Correct it to "host-derived". Also, the parenthesis is cutting the flow and it is redundant with the main text. Please rewrite it.
- Line 36 - The authors start the "importance" section repeating what they just said in the previous section "the microbiome (Symbiodiniaceae and bacteria) in the CWCs 37 from depth down to 260-370 m were illustrated". Remove the whole sentence.
- Line 37 - Remove "Importantly", it is unnecessary.
- Line 40-41 - "Our present results indicated that the energy requirements of CWCs were mainly acquired by 41 bacteria via non-photosynthetic CO₂ fixation pathway". This sentence is not proven, the results do not indicate this. It is too much speculation.

- Line 43-45 - The last sentence can be improved to become clearer.

Introduction

The introduction is confusing and does not follow a clear flow. The information is relevant, but the ideas are disconnected and loose. It needs to be reviewed and rewritten.

- Line 48-50 - What about ROV? They are a huge part of these advancements.

- Line 48-59 - The first paragraph has relevant information but all the sentences are disconnected and/or out of place, breaking the flow and making the text confusing. Please, improve the whole paragraph.

Line 60 - Change "a range of microorganisms" to "different groups of microorganisms".

Line 63 - Change "focus" to "focused".

Line 64 - Change "illustrated" to "shown".

Line 65 - "that THE ASSOCIATED microbiome".

Line 67-68 - Change "In WWCs, bacteria has been implicated in several other services" to "Additionally, bacterial communities associated with WWCs have been proved to play several important roles, such..."

Line 69-73 - These two sentences are completely out of place. The authors simply go from the importance of the associated microbiome to bleaching and disease without explaining much about these subjects. These are both very complex topics that should be a little more exploited (if mentioned).

Line 74-77 - Again, two sentences completely disconnected between themselves and, also, disconnected from the previous paragraph.

Line 102 - "from depths down to 260-370 m" Same concern mentioned in the abstract section

Line 102-103 - "to address THE questions mentioned above"

Line 105 - Remove "commonly known".

Material and Methods

It is not very clear how many samples of each coral species were used for each analysis. For example, how many replicates (not pseudoreplicates) were used for the molecular results. Please, make it clearer in the section.

Line 303-304 - Change "during a cruise in the eastern (112.918-112.934{degree sign} E, 304 18.741-18.764{degree sign} N) Hainan Island of the South China Sea on December of 2021" to "during a cruise in the eastern Hainan Island of the South China Sea (112.918-112.934{degree sign} E, 304 18.741-18.764{degree sign} N) on December of 2021"

Line 305 - "For each CWC, THE colony was 306 pressing down..."

Line 306-307 - Change to "Branches (about 5-10 cm in 307 height) were collected from the coral colonies using the submersible's manipulator claw"

Line 307 - Same as above.

Line 310-311 - Change to "were collected from, Sanya, Hainan Island, China (109.292{degree sign} E, 18.123{degree sign} N), a tropical coral reef, by snorkeling..."

Line 317 - Change "that contains" to "containing"

Line 323 - Change "are" to "were". Please, be consistent with the language, if you choose past tense, keep everything in past tense.

Results

Major concern:

The authors start the section by saying that "Due to the difficulty in obtaining replicate samples of each CWC species, we mainly focused on 112 the description of different taxon of Symbiodiniaceae and bacteria in explored corals". However, instead of simply describing the taxa, they start talking about "abundance". I don't think this should be even mentioned. If there are not enough replicates to calculate a reliable "abundance", simply keep the description of the taxa.

Also, I would change the analysis from OTUs to ASVs.

Minor:

Line 112 - Change "taxon" to "taxa" and "explored" to "sampled"

Discussion:

Line 215 - Remove the sentence "The present results showed CWCs from depth of 260-370 m hosted Symbiodiniaceae."

Line 229 - Change "Importantly, we firstly observed binary fission cells of 229 Symbiodiniaceae in the endoderm tissues of the CWCs living in deep-waters." To "Importantly, we reported binary fission cells of 229 Symbiodiniaceae in the endoderm tissues of the CWCs living in deep-waters for the first time."

Line 232-234 - "These results indicated that the Symbiodiniaceae could be a symbiont or parasite of 233 CWCs and had cellular activity, which were important for Symbiodiniaceae acclimation to deep234 water inhabitants" I disagree with this conclusion, I think it is too much speculation. The results do not suggest this.

Line 249 - Change "supporting" to "supports" and "speculate" to "speculations"?

Line 250 - "Furthermore..." Should be a new paragraph.

Line 259 - "nitrogen cycles" to "nitrogen cycle"

Summary

Line 285-286 - Change "the microbiome (Symbiodiniaceae and bacteria) in the CWC from depth down to 286 200-300 m were

illustrated." To "the microbiome (Symbiodiniaceae and bacteria) associated with CWC from depth ranging from 200-300 m (this is different from the rest...) were investigated."

Line 288-289 - "The energy requirements of CWCs were 289 mainly acquired by bacteria via a non-photosynthetic CO₂ fixation pathway." The results do not support this sentence. It should be removed or reformulated as merely a speculation.

Staff Comments:

Preparing Revision Guidelines

Please return the manuscript within 60 days; if you cannot complete the modification within this time period, please contact me. If you do not wish to modify the manuscript and prefer to submit it to another journal, please notify me of your decision immediately so that the manuscript may be formally withdrawn from consideration by Microbiology Spectrum.

Corrections to the manuscript

Dear Editors and Reviewers,

We appreciate the time and efforts by the editors and reviewers in reviewing the manuscript entitled “Unfolding the Secrets of Microbiome (Symbiodiniaceae and Bacteria) In Cold-water Coral”(Manuscript ID: Microbiology Spectrum - Spectrum01315-23). The newly submitted manuscript has made point-to-point revision according to the editor’s and reviewer’s comments. In the revised manuscript, all corrections have been marked in red. The detailed responses and corrections are listed below.

Reviewer #1

(Comments for the Author):

In this study, Gong et al. analyzed the microbiome of cold-water coral. they found that CWC host low-abundance Symbiodiniaceae and predominantly of diverse communities of bacteria. Importantly, their present results indicated that Symbiodiniaceae might be a symbiont or parasite of CWC and has cellular activity, these finds were important for photosynthetic algae to acclimation to deep water habitats. In addition, this study revealed that bacteria were abundant in CWC, which exhibited a high-level transcription of genes for carbon fixation via the Wood-Ljungdahl pathway, as well as short chain fatty acids (acetate, butyrate and propionate) production, nitrogen and sulfur (especially host derived organic nitrogen and sulfur metabolisms) cycles, suggesting the energy requirements of CWC were mainly acquired by bacteria via non-photosynthetic CO₂ fixation pathway. The findings advanced general understanding of the mechanisms of CWCs surviving in deep-water environments, which which receive less than 1% of surface irradiance. In all, this is an interesting study. However, some minor revision is needed that may improve the quality of the manuscript.

Response: Thanks for the meticulous checking and helpful suggestions.

1. Line 37: were should be revised to was

Response: Thanks for the meticulous checking and helpful suggestions. Due to repeat with previous section, the whole sentence has been deleted in revised manuscript. The singular and plural of verbs have been checked throughout the manuscript.

2. Line 41: CO₂ should be revised to CO₂.

Response: Thanks for the meticulous checking and helpful suggestions. The “CO₂” has been revised to “CO₂” throughout the manuscript.

3. Line 74: There is a problem of the reference format (Hourigan., 2007), please check and revise

Response: Thanks for the meticulous checking and helpful suggestions. The reference citation has been checked and revised.

4. Line 75: "been inferred" been should be deleted.

Response: Thanks for the meticulous checking and helpful suggestions. “been” has been deleted in revised manuscript.

5. Line 76: There is a problem of the reference format (Hourigan., 2007), please check and revise

Response: Thanks for the meticulous checking and helpful suggestions. The reference citation has been checked and revised.

6. Line 77: for should be revised to on

Response: Thanks for the meticulous checking and helpful suggestions. The “for” has been revised into “on” in newly submitted manuscript.

7. Line 83: There is a problem of the reference format (Hourigan., 2007), please check and revise

Response: Thanks for the meticulous checking and helpful suggestions. Reference citation has been checked and revised.

8. Line 94: less is revised to less are

Response: Thanks for the meticulous checking and helpful suggestions. The original sentence “While these CWCs, more specifically *Leptoseris* spp., have received a lot of attention, comparatively less is understood about cold-water octocoral species” has

been changed to “While attention has been focused on *Leptoseris* spp., there is comparatively less knowledge on cold-water octocoral species” in revised manuscript.

9. Line 156: Carbohydrates should be revised to carbohydrates

Response: Thanks for the meticulous checking and helpful suggestions. The word “Carbohydrates” has been revised to “carbohydrates”.

10. Line 178: Nitrogen cycle should be revised to nitrogen cycle

Response: Thanks for the meticulous checking and helpful suggestions. Nitrogen cycle has been revised to nitrogen cycle.

11. Line 179: of nitrogen cycle should be revised to for nitrogen cycle

Response: Thanks for the meticulous checking and helpful suggestions. Related sentence “An increased potential of nitrogen cycle, including nitrogen transport, inorganic nitrogen fixation, nitrification and denitrification related processes, was observed in bacteria of CWCs” has been changed to “Bacteria associated with CWC were found to have an increased potential for nitrogen cycle processes, including nitrogen transport, inorganic nitrogen fixation, nitrification and denitrification” in revised manuscript.

12. Line 182: were primarily assigned should be revised to was primarily assigned

Response: Thanks for the meticulous checking and helpful suggestions. Related sentence “The increased relative abundance of three genes encoding nitrogen fixation protein (nifB, >0.5-times, nifT, >1-times and nifU, 8-times) were primarily assigned to bacteria associated with CWCs” has been changed to “The relative abundance of three genes encoding nitrogen fixation protein (nifB, >0.5-times, nifT, >1-times and nifU, 8-times) was significantly higher in CWC-associated bacteria ” in revised manuscript.

13. Line 186: an notable should be revised to a notable

Response: Thanks for the meticulous checking and helpful suggestions. Related sentences “Simultaneously, the bacteria associated with CWCs exhibited a notable increased potential for urea transport and degradation, as well as glutamine and glutamate biosynthesis, an increase in genes encoding urea transporter (63-times,

urtA/B/C/D), urease subunits and accessory proteins (640-times on average), glutamine synthetase, which assimilates one molecule each of ammonia, glutamate and ATP into glutamine (1518-times), and glutamate synthase, which leads to glutamate (546-times, *gltB/D*)." have been changed to "Furthermore, the bacteria associated with CWC exhibited an escalated potential for urea transport and degradation, as well as glutamine and glutamate biosynthesis. This was evidenced by an increase in genes encoding urea transporter (63-times, *urtA/B/C/D*), urease subunits and accessory proteins (640-times on average), glutamine synthetase (1518-times, *glnA*) and glutamate synthase (546-times, *gltB/D*)." in revised manuscript.

14. Line 195: Sulfur cycle should be revised to sulfur cycle

Response: Thanks for the meticulous checking and helpful suggestions. Sulfur cycle has be revised to sulfur cycle in revised manuscript.

15. Line 244-245: the sentence "indicating non-photosynthetic CO₂ fixation by bacteria is play important roles in organic carbon supply of CWCs." Should be revised to "indicating non-photosynthetic CO₂ fixation by bacteria is playing important roles in supplying organic carbon to CWCs."

Response: Thanks for the meticulous checking and helpful suggestions. related description has been revised into "indicating non-photosynthetic CO₂ fixation by bacteria is playing important roles in supplying organic carbon to CWC." according to your suggestion.

16. Line 246: the word "acquire" should be revised to "acquired"

Response: Thanks for the meticulous checking and helpful suggestions. the word "acquire" has be changed to "acquired" in revised manuscript.

17. Line 251: the word "were" should be revised to "was"

Response: Thanks for the meticulous checking and helpful suggestions. The singular and plural of verbs have been checked throughout the manuscript.

18. Line 289: the word "were" should be revised to "was"

Response: Thanks for the meticulous checking and helpful suggestions. The singular and plural of verbs have been checked throughout the manuscript.

19. Line 330: In details should be revised to In detail

Response: Thanks for the meticulous checking and helpful suggestions. In details has been revised to In detail.

20. Line 378: the "a" should be revised to an

Response: Thanks for the meticulous checking and helpful suggestions. a has been changed to an.

21. FIG 2. A and B cannot be found, please check.

Response: Thanks for the meticulous checking and helpful suggestions. Related description has been checked and revised.

22. FIG 3. A and B cannot be found, please check.

Response: Thanks for the meticulous checking and helpful suggestions. Related description has been checked and revised.

Thanks for the meticulous checking and helpful suggestions again.

Reviewer #2 (Comments for the Author):

1. Major concerns:

- The paper is relevant to the field, showing important findings that will help to fill gaps in the area. However, the number of replicates (or even pseudoreplicates, it is still unclear) makes me think that the whole results and discussion sections will need to be reformulated. If the authors are using pseudoreplicates (as it seems like), they should not do any comparison and not work with abundance analyses but simply a description of the findings. Also, the statistics should be reviewed if this is the case. Once the authors make the methodology clear regarding the number of true replicates they used for each analysis, we can better judge what can be done to show and discuss

the results.

Response: Thanks for the meticulous checking and helpful suggestions. In reality, the number of replicates in this study is pseudoreplicates. Description of the number of pseudoreplicates is added in MATERIALS AND METHODS part of revised manuscript. “a total of three CWC samples were used in this study, which including three different CWC species, *N. versluysi*, *H. uatumani* and *Muriceides* sp.” For each coral species, one sample was used for related analysis due to obtaining replicate samples of each CWC species challenged in deep-waters. Therefore, the results and discussion parts were majorly focused on description of findings in revised manuscript.

2. The paper needs English review AND technical review.

Response: Thanks for the meticulous checking and helpful suggestions. Related paragraphs and sentences have been revised according to your suggestions. and the revised manuscript has been corrected by an native English speaker.

Abstract

3. Line 20 - Change "habitants" for "habitats"? It is unclear.

Response: Thanks for the meticulous checking and helpful suggestions. the “habitants” has been changed into “habitats” throughout the manuscript.

4. Line 20 - 21 - "Detected microbiome" does not sound right. Consider changing it to "In this study, we investigated the CWC-associated microbiome...".

Response: Thanks for the meticulous checking and helpful suggestions. Related sentences “In this study, we detected microbiome (Symbiodiniaceae and bacteria) using metabarcoding and tissue section observation, and screened the metatranscriptomes of sampled CWCs (*Narella versluysi*, *Heterogorgia uatumani* and *Muriceides* sp.) from depths down to 260-370 m.” have been changed into “This

study utilized metabarcoding, tissue section observation and metatranscriptomes to investigate the microbiome (Symbiodiniaceae and bacteria) of CWC species (*Narella versluysi*, *Heterogorgia uatumani* and *Muriceides* sp.) from depths ranging from 260 m to 370 m.”. in revised manuscript.

5. Line 22- The sentence is too long and has grammar mistakes. Consider changing it to "We also analyzed the metatranscriptomes of THE sampled CWCs...".

Response: Thanks for the meticulous checking and helpful suggestions. Related sentences “In this study, we detected microbiome (Symbiodiniaceae and bacteria) using metabarcoding and tissue section observation, and screened the metatranscriptomes of sampled CWCs (*Narella versluysi*, *Heterogorgia uatumani* and *Muriceides* sp.) from depths down to 260-370 m.” have been changed into “This study utilized metabarcoding, tissue section observation and metatranscriptomes to investigate the microbiome (Symbiodiniaceae and bacteria) of CWC species (*Narella versluysi*, *Heterogorgia uatumani* and *Muriceides* sp.) from depths ranging from 260 m to 370 m.” in revised manuscript.

6. Line 23 - "from depths down to 260-370 m" Is it the total range or the range of the deepest spot? It is confusing.

Response: Thanks for the meticulous checking and helpful suggestions. “from depths down to 260-370 m” has been revised into “from depths ranging from 260 m to 370 m” in corrected manuscript.

7. Line 24 - It is plural, so it should be "were used as control groups".

Response: Thanks for the meticulous checking and helpful suggestions. Related description has been changed into “were used as control groups”

8. Line 25-26 - "and predominantly of diverse communities of bacteria" The sentence is confusing. Please rewrite it.

Response: Thanks for the meticulous checking and helpful suggestions. Related description has been revised into "Results revealed that CWC host diverse bacteria and Symbiodiniaceae cells were observed in endoderm of CWC tissues."

9. Line 26-27 - "We designated several new candidate bacterial phyla" Grammar is wrong here and the word designated is not appropriate for what the authors mean.

Response: Thanks for the meticulous checking and helpful suggestions. Related description has been changed into "Several new candidate bacterial phyla were found in both CWC and WWC, including Coralsanbacteria, Coralqiangbacteria, Coralgsqaceae, Coralgongineae and etc."

10. Line 28 - Remove "and/or", it should be "and" only.

Response: Thanks for the meticulous checking and helpful suggestions. "and/or" has been revised into "and".

11. Line 29 - Consider changing "16S metabarcoding sequencing" to "the 16S rRNA gene sequencing".

Response: Thanks for the meticulous checking and helpful suggestions. "16S metabarcoding sequencing" has been revised into "the 16S rRNA gene sequencing".

12 Line 30-32 - Consider changing it to "we noted that the coral-associated Symbiodiniaceae community showed a low-level transcription of genes involved in photosynthesis, CO₂ fixation, glycolysis, citric acid cycle and other core function pathways in the CWCs."

Response: Thanks for the meticulous checking and helpful suggestions. Related sentences “At the gene transcription level, we noted that the Symbiodiniaceae showed a low-level transcription of genes involved in photosynthesis, CO₂ fixation, glycolysis, citric acid cycle and other core function pathways in the CWCs.” have been changed into “At the gene transcription level, the CWC-associated Symbiodiniaceae community showed a low-level transcription of genes involved in photosynthesis, CO₂ fixation, glycolysis, citric acid cycle.....”

13. Line 35 - Correct it to "host-derived". Also, the parenthesis is cutting the flow and it is redundant with the main text. Please rewrite it.

Response: Thanks for the meticulous checking and helpful suggestions. Related sentences “The bacteria exhibited a high-level transcription of genes for carbon fixation via the Wood-Ljungdahl pathway, as well as short chain fatty acids (acetate, butyrate and propionate) production, nitrogen and sulfur (especially host derived organic nitrogen and sulfur metabolisms) cycles in the CWCs.” have been changed into “bacteria associated with CWC exhibited a high-level transcription of genes for carbon fixation via the Wood-Ljungdahl pathway, short chain fatty acids production, nitrogen and sulfur cycles.”

14 Line 36 - The authors start the "importance" section repeating what they just said in the previous section "the microbiome (Symbiodiniaceae and bacteria) in the CWCs 37 from depth down to 260-370 m were illustrated". Remove the whole sentence.

Response: Thanks for the meticulous checking and helpful suggestions. the whole sentence has been deleted according to your suggestion in revised manuscript.

15. Line 37 - Remove "Importantly", it is unnecessary.

Response: Thanks for the meticulous checking and helpful suggestions. “Importantly” has been deleted in revised manuscript.

16. Line 40-41 - "Our present results indicated that the energy requirements of CWCs were mainly acquired by 41 bacteria via non-photosynthetic CO₂ fixation pathway". This sentence is not proven, the results do not indicate this. It is too much speculation.

Response: Thanks for the meticulous checking and helpful suggestions. Related sentence “Our present results indicated that the energy requirements of CWCs were mainly acquired by bacteria via non-photosynthetic CO₂ fixation pathway.” has been changed into “This study also revealed complete non-photosynthetic CO₂ fixation pathway of bacteria in CWC” in revised manuscript.

17 Line 43-45 - The last sentence can be improved to become clearer.

Response: Thanks for the meticulous checking and helpful suggestions. The last sentence has been improved into “These findings highlight the important role of bacteria in the carbon, nitrogen sulfur cycles in CWC, which were possibly crucial for CWC survival in in deep-water environments ”.

Introduction

18. The introduction is confusing and does not follow a clear flow. The information is relevant, but the ideas are disconnected and loose. It needs to be reviewed and rewritten.

Response: Thanks for the meticulous checking and helpful suggestions. The introduction part has been rewritten in revised manuscript.

19 Line 48-50 - What about ROV? They are a huge part of these advancements.

Response: Thanks for the meticulous checking and helpful suggestions. Related

description has been changed into “However, only recent advancements in acoustic survey techniques and submersible tools (such as Remotely-Operated Vehicle)” according to your suggestion.

20 Line 48-59 - The first paragraph has relevant information but all the sentences are disconnected and/or out of place, breaking the flow and making the text confusing. Please, improve the whole paragraph.

Response: Thanks for the meticulous checking and helpful suggestions. this paragraph has been rewritten in revised manuscript.

21. Line 60 - Change "a range of microorganisms" to "different groups of microorganisms".

Response: Thanks for the meticulous checking and helpful suggestions. “a range of microorganisms” has been changed into “ different groups of microorganisms” according to your suggestion.

22. Line 63 - Change "focus" to "focused".

Response: Thanks for the meticulous checking and helpful suggestions. Related sentences have been changed into “While the microboime of the most commonly studied reef-building coral in warm-water habitats (WWC) has been extensively researched (Gong *et al.*, 2018; van de Water *et al.*, 2018; Bollati *et al.*, 2020; Gong, Jin, Ren, *et al.*, 2020), there has been limited attention given to the microbiome of CWC due to cost and difficulty in sample retrieval (Hourigan., 2007).”

23. Line 64 - Change "illustrated" to "shown".

Response: Thanks for the meticulous checking and helpful suggestions. Related

sentence has been changed into “Both Symbiodiniaceae and bacteria associated with WWC play active roles in the health and adaptive response of the host to environmental changes (Morris *et al.*, 2019; Apprill, 2020).”

24. Line 65 - "that THE ASSOCIATED microbiome".

Response: Thanks for the meticulous checking and helpful suggestions. Related sentence has been removed and revised into the sentence as described above (23).

25. Line 67-68 - Change "In WWCs, bacteria has been implicated in several other services" to "Additionally, bacterial communities associated with WWCs have been proved to play several important roles, such..."

Response: Thanks for the meticulous checking and helpful suggestions. Related sentence has been changed into “Bacterial communities associated with WWC has been proved to play several important roles, such as nitrogen fixation, sulfur cycling, antibiotic production (Gong, Jin, Ren, *et al.*, 2020; Tandon *et al.*, 2020).”

26. Line 69-73 - These two sentences are completely out of place. The authors simply go from the importance of the associated microbiome to bleaching and disease without explaining much about these subjects. These are both very complex topics that should be a little more exploited (if mentioned).

Response: Thanks for the meticulous checking and helpful suggestions. Related sentences has been deleted in revised manuscript.

27. Line 74-77 - Again, two sentences completely disconnected between themselves and, also, disconnected from the previous paragraph.

Response: Thanks for the meticulous checking and helpful suggestions. Thanks for

the meticulous checking and helpful suggestions. Related sentences has been deleted in revised manuscript.

28. Line 102 - "from depths down to 260-370 m" Same concern mentioned in the abstract section

Response: Thanks for the meticulous checking and helpful suggestions. "from depths down to 260-370 m" has been revised into "from depths ranging from 260 m to 370 m"

29. Line 102-103 - "to address THE questions mentioned above"

Response: Thanks for the meticulous checking and helpful suggestions. Related description has been removed in revised manuscript.

30. Line 105 - Remove "commonly known".

Response: Thanks for the meticulous checking and helpful suggestions. "commonly known" has been removed in newly submitted manuscript.

Material and Methods

31. It is not very clear how many samples of each coral species were used for each analysis. For example, how many replicates (not pseudoreplicates) were used for the molecular results. Please, make it clearer in the section.

Description of the number of pseudoreplicates is added in MATERIALS AND METHODS part of revised manuscript. "a total of three CWC samples were used in this study, which including three different CWC species, *N. versluyisi*, *H. uatumani* and *Muriceides* sp." For each coral species, one sample were used for related analysis due to obtaining replicate samples of each CWC species challenged in deep-waters. Therefore, the results and discussion parts were majorly focused on description of

findings in revised manuscript.

32. Line 303-304 - Change "during a cruise in the eastern (112.918-112.934{degree sign} E, 304 18.741-18.764{degree sign} N) Hainan Island of the South China Sea on December of 2021" to "during a cruise in the eastern Hainan Island of the South China Sea (112.918-112.934{degree sign} E, 304 18.741-18.764{degree sign} N) on December of 2021

Response: Thanks for the meticulous checking and helpful suggestions. Related sentences have been revised into “During a cruise in the eastern Hainan Island of the South China Sea (112.918-112.934° E, 18.741-18.764° N) on December of 2021, a total of three CWC samples (*N. versluyisi*, *H. uatumani* and *Muriceides* sp.) were collected by using a submersible (SHEN HAI YONG SHI).”

33. Line 305 - "For each CWC, THE colony was 306 pressing down..."

Response: Thanks for the meticulous checking and helpful suggestions. Related description has been changed into “For each CWC, the colony was pressing down...”.

34. Line 306-307 - Change to "Branches (about 5-10 cm in 307 height) were collected from the coral colonies using the submersible's manipulator claw"

Response: Thanks for the meticulous checking and helpful suggestions. Related description has been revised into “Branches (about 5-10 cm in height) were collected from from CWC colonies using the submersible’s manipulator claw” according to your suggestion.

35. Line 307 - Same as above.

Response: Thanks for the meticulous checking and helpful suggestions. Related

description has been revised as above.

36. Line 310-311 - Change to "were collected from, Sanya, Hainan Island, China (109.292{degree sign} E, 18.123{degree sign} N), a tropical coral reef, by snorkeling..."

Response: Thanks for the meticulous checking and helpful suggestions. Related sentence has been changed into "A total of three WWC samples (*Acropora* sp., *G. fascicularis* and *P. lamellina*) were collected from, Sanya, Hainan Island, China (109.292° E, 18.123° N), a tropical coral reef, by snorkeling on December of 2021." according to your suggestion.

37. Line 317 - Change "that contains" to "containing"

Response: Thanks for the meticulous checking and helpful suggestions. "that contains" has been changed into "containing" in newly submitted manuscript.

38. Line 323 - Change "are" to "were". Please, be consistent with the language, if you choose past tense, keep everything in past tense.

Response: Thanks for the meticulous checking and helpful suggestions. the "are" has been changed into "were", and the tense has been checked throughout the manuscript.

Results

39. Major concern:

The authors start the section by saying that "Due to the difficulty in obtaining replicate samples of each CWC species, we mainly focused on 112 the description of different taxon of Symbiodiniaceae and bacteria in explored corals". However, instead of simply describing the taxa, they start talking about "abundance". I don't think this

should be even mentioned. If there are not enough replicates to calculate a reliable "abundance", simply keep the description of the taxa.

Response: Thanks for the meticulous checking and helpful suggestions. due to limited samples, the results and discussion parts were majorly focused on description of findings in revised manuscript.

40. Also, I would change the analysis from OTUs to ASVs.

Response: Thanks for the meticulous checking and helpful suggestions. "OTUs" has been changed into "ASVs".

Minor:

41. Line 112 - Change "taxon" to "taxa" and "explored" to "sampled"

Response: Thanks for the meticulous checking and helpful suggestions. "taxon" has been changed into "taxa", and "explored" has been changed into "sampled".

Discussion:

42. Line 215 - Remove the sentence "The present results showed CWCs from depth of 260-370 m hosted Symbiodiniaceae."

Response: Thanks for the meticulous checking and helpful suggestions. This sentence has been deleted in revised manuscript.

43. Line 229 - Change "Importantly, we firstly observed binary fission cells of 229 Symbiodiniaceae in the endoderm tissues of the CWCs living in deep-waters." To "Importantly, we reported binary fission cells of 229 Symbiodiniaceae in the endoderm tissues of the CWCs living in deep-waters for the first time."

Response: Thanks for the meticulous checking and helpful suggestions. Related

sentence has been revised according to your suggestion in newly submitted manuscript.

44. Line 232-234 - "These results indicated that the Symbiodiniaceae could be a symbiont or parasite of 233 CWCs and had cellular activity, which were important for Symbiodiniaceae acclimation to deep²³⁴ water habitats" I disagree with this conclusion, I think it is too much speculation. The results do not suggest this.

Response: Thanks for the meticulous checking and helpful suggestions. I agree with your idea and related description has been changed into "These results demonstrated that Symbiodiniaceae can survive and actively transcribe genes in CWC, suggesting a possible symbiotic or parasitic relationship with the host."

45. Line 249 - Change "supporting" to "supports" and "speculate" to "speculations"?

Response: Thanks for the meticulous checking and helpful suggestions. "supporting" has been changed into "supports", and "speculate" has been changed into "speculations". in newly submitted manuscript.

46. Line 250 - "Furthermore..." Should be a new paragraph.

Response: Thanks for the meticulous checking and helpful suggestions. Related sentences were splited into a new paragraph according to your suggestion.

47. Line 259 - "nitrogen cycles" to "nitrogen cycle"

Response: Thanks for the meticulous checking and helpful suggestions. "nitrogen cycles" has been revised into "nitrogen cycle".

Summary

48. Line 285-286 - Change "the microbiome (Symbiodiniaceae and bacteria) in the CWC from depth down to 286 200-300 m were illustrated." To "the microbiome (Symbiodiniaceae and bacteria) associated with CWC from depth ranging from

200-300 m (this is different from the rest...) were investigated."

Response: Thanks for the meticulous checking and helpful suggestions. Related sentence has been changed into "the microbiomes (Symbiodiniaceae and bacteria) associated with CWC from depth ranging from 200-300 m were investigated." according to your suggestion.

49. Line 288-289 - "The energy requirements of CWCs were 289 mainly acquired by bacteria via a non-photosynthetic CO₂ fixation pathway." The results do not support this sentence. It should be removed or reformulated as merely a speculation.

Response: Thanks for the meticulous checking and helpful suggestions. Related sentence has been changed into "The bacteria associated with CWC exhibited transcription of genes for the Wood-Ljungdahl pathway, suggesting that the energy requirements of CWC were possibly acquired by bacteria via a non-photosynthetic CO₂ fixation pathway in deep-water.

Thanks for the meticulous checking and helpful suggestions again.

Looking forward to hearing from you and with best wishes.

Sincerely yours,

Prof. Dr. Meixia Zhao

Prof. Dr. Kefu Yu

Dr. Sanqiang Gong

July 9, 2023

Dr. Sanqiang Gong
South China Sea Institute of Oceanology, Chinese Academy of Sciences
Guangzhou, China
Guangzhou
China

Re: Spectrum01315-23R1 (Unfolding the secrets of microbiome (Symbiodiniaceae and bacteria) in cold-water coral)

Dear Dr. Sanqiang Gong:

Your manuscript has been accepted, and I am forwarding it to the ASM Journals Department for publication. You will be notified when your proofs are ready to be viewed.

Sincerely,

Konstantinos Kormas
Editor, Microbiology Spectrum
